# Early postoperative complications following laparoscopic-assisted modified Soave procedure for Hirschsprung's disease: Incidence, Clavien-Dindo classification, and risk factor analysis

Xianhui Shang[1,2☯], Renbiao Huang[3☯], Yuanmei Liu[1,2☯*], Hao Fu[1,2], Luping Xiang[1,2], Shiyu Xu[1,2], Yan Qu[4]

**1** Department of Pediatric Surgery, Affiliated Hospital of Zunyi Medical University, Zunyi, China,
**2** Department of Pediatric Surgery, Guizhou Children's Hospital, Zunyi, China, **3** Department of Pediatric Surgery, Shenzhen Hospital, University of Chinese Academy of Sciences, Shenzhen, China, **4** Department of Gastrointestinal Surgery, The Second Affiliated Hospital of Zunyi Medical University, Zunyi, China

☯ These authors contributed equally to this work.
* yuanmei116@aliyun.com

## Abstract

### Objective

To investigate the incidence and severity of early postoperative complications (within 30 days post-surgery) following laparoscopic-assisted modified Soave procedure for Hirschsprung's disease (HSCR), and to analyze the factors influencing these complications.

### Methods

A retrospective analysis was performed on the clinical data of patients who underwent laparoscopic-assisted modified Soave procedure for HSCR from January 2010 to December 2020 at our institution. Data including gender, age at surgery, clinical type, occurrence of Hirschsprung-associated enterocolitis (HAEC), presence of stoma, anemia, hypoalbuminemia, and surgery duration were collected. Patients were categorized into complication and non-complication groups based on the occurrence of early postoperative complications. The Clavien-Dindo (CD) classification system was used to grade the severity of postoperative complications, and univariate analysis was performed to identify potential factors influencing complications. Statistically significant variables were further analyzed by multivariate logistic regression.

### Results

A total of 112 patients were included in the study, comprising 83 males (74.1%) and 29 females (25.9%). Nineteen patients (17.0%) experienced complications, including three with two types of complications. The non-complication group included 93

**Data availability statement:** All relevant data are within the manuscript and its Supporting Information files.

**Funding:** This work was supported by the Guizhou Provincial Health Commission (Grant No. gzwkj2024-431, received by Xianhui Shang), the Guizhou Provincial Administration of Traditional Chinese Medicine (Grant No. QZYY-2024-015, received by Xianhui Shang), and the Zunyi Science and Technology Planning Project (Grant No. Zunyi Kehe HZ [2023] No. 145, received by Yan Qu). The funders had no role in study design, data collection and analysis, decision to publish, or preparation of the manuscript.

**Competing interests:** The authors have declared that no competing interests exist.

patients (83.0%). The complications included perianal skin erosion (2 cases, 9.1%), abdominal wound infection (1 case, 4.6%), abdominal cavity infection (1 case, 4.6%), HAEC (10 cases, 45.5%), HAEC with intestinal perforation (2 cases, 9.1%), umbilical wound dehiscence with omental exposure (1 case, 4.6%), adhesive bowel obstruction (3 cases, 13.6%), rectal retraction (1 case, 4.5%), and acute respiratory failure with laryngeal edema (1 case, 4.6%). The CD classification for postoperative complications was as follows: Grade I (3 cases, 13.6%), Grade II (11 cases, 50.0%), Grade IIIb (7 cases, 31.8%), and Grade IVa (1 case, 4.6%). Univariate analysis indicated that preoperative stoma ($P=0.029$), preoperative anemia ($P=0.025$), and hypoalbuminemia ($P<0.001$) were significant factors influencing early postoperative complications. Multivariate logistic regression revealed that preoperative stoma (OR=4.826, 95% CI = 1.187–20.162, $P=0.028$) and hypoalbuminemia (OR=9.14, 95% CI = 2.678–30.972, $P<0.001$) were independent risk factors for postoperative complications.

## Conclusion

Early postoperative complications following laparoscopic-assisted modified Soave procedure are not uncommon, with approximately one-third of cases requiring surgical reintervention. The most frequent complication was Hirschsprung-associated enterocolitis (HAEC), and most were classified as Clavien-Dindo grade II. Preoperative hypoalbuminemia and stoma creation were identified as independent risk factors, suggesting that perioperative nutritional intervention and risk stratification may help reduce the incidence of complications.

---

## Introduction

Hirschsprung's disease (HSCR) is a common congenital gastrointestinal malformation in pediatric patients. Due to the complex etiology, genetics, pathology, and pathophysiology of HSCR, the incidence of postoperative complications such as recurrent constipation, small bowel colitis, and fecal soiling remains relatively high [1]. Since 2002, our institution has employed a modified Soave procedure with stepwise transanal resection of the rectal muscle cuff, which has yielded favorable outcomes [2]; however, some complications persist. Although numerous reports on postoperative complications of HSCR exist, most of the studies are based on experiences from large, single-center hospitals or specific surgical treatment groups, and there is no universally standardized, objective criterion for evaluation.

To address this gap, our study introduces a systematic application of the Clavien-Dindo (CD) classification system to grade early postoperative complications (≤30 days) following laparoscopic-assisted modified Soave procedures in children with HSCR. While the CD system has been widely used in adult surgery, its structured use in pediatric HSCR patients remains extremely limited in both domestic and international literature [3].

Furthermore, we conducted multivariate logistic regression and identified preoperative hypoalbuminemia (albumin <35 g/L) as a novel independent risk factor for postoperative complications—an association not previously established in HSCR-specific studies.

This retrospective analysis includes 112 patients treated from 2010 to 2020, representing a relatively large sample size with a comprehensive data structure including surgical type, perioperative variables, complication types, and CD grades.

Our findings aim not only to standardize complication reporting, but also to provide clinically actionable insight—highlighting the importance of preoperative nutritional optimization, particularly the correction of hypoalbuminemia and anemia, to improve surgical outcomes.

The use of standardized diagnostic and grading systems for complications would facilitate comparative studies between centers and provide a better assessment of surgical outcomes. Therefore, this study retrospectively analyzed the clinical data of patients who underwent laparoscopic-assisted modified Soave procedure for HSCR at the Department of Pediatric Surgery, Zunyi Medical University Affiliated Hospital, from January 2010 to December 2020. The Clavien-Dindo (CD) classification system was employed to assess early postoperative complications (≤30 days) [4], and the risk factors for these complications were analyzed. Internationally comparable studies have been included to contextualize our results, complementing domestic data and enhancing the relevance of this research.

Together, these contributions aim to fill methodological gaps in HSCR surgical literature and support evidence-based perioperative strategies.

## Materials and methods

### 1. Study subjects

A retrospective analysis was conducted on the clinical data of patients who underwent laparoscopic-assisted modified Soave procedure at our institution between January 2010 and December 2020. The data used in this study were accessed from the Medical Records Management Database of Zunyi Medical University Affiliated Hospital on December 1, 2024. The study was approved by the Ethics Committee of Zunyi Medical University Affiliated Hospital (KLLY-2020–039). Informed consent was obtained from all patients' families. For some patients, a temporary preoperative stoma was performed. The main indications included: (1) recurrent episodes of severe Hirschsprung-associated enterocolitis (HAEC) that were refractory to conservative treatment; (2) poor preoperative nutritional status, such as significant hypoalbuminemia or anemia, that could not be corrected in a short time; and (3) imaging findings suggestive of long segment or total colonic aganglionosis, requiring further clarification of the disease extent to ensure the safety and efficacy of definitive surgery.

#### Inclusion criteria.

1. Patients with a clinical history, physical examination, and auxiliary examinations consistent with Hirschsprung's disease (HSCR), who received laparoscopic-assisted modified Soave procedure;

2. Diagnosis of short segment or long segment HSCR based on preoperative barium enema results and intraoperative frozen section pathology;

3. Patients who developed early postoperative complications (≤30 days) following HSCR surgery;

4. Patients with complete clinical data.

#### Exclusion criteria.

1. Patients with congenital diseases other than HSCR;

2. Patients with megacolon of other etiologies;

3. Patients with total colonic or total intestinal HSCR;

4. HSCR patients with incomplete clinical data affecting postoperative complication analysis.

## 2. Surgical technique

After endotracheal intubation and general anesthesia, a conventional three-port laparoscopic approach was used. The thickness, peristalsis, and other characteristics of the bowel were observed to identify the areas of stenosis, transition, and dilation. Full-thickness bowel wall biopsies were taken from the stenotic and anastomotic segments for frozen section analysis to determine the extent of resection. The mesorectum and rectal lateral ligaments were separated using an ultrasonic scalpel up to 1–2 cm below the peritoneal reflection. The proximal bowel was freed to normal colon. The perineal procedure was as follows: a trapezoidal incision was made with a needle-type electrocautery approximately 0.5 cm from the posterior anal verge and 0.8 cm from the anterior anal verge, cutting through the rectal mucosa. A traction suture of 0 silk thread was placed around the mucosal incision. The mucosa was separated layer by layer and advanced 1–1.5 cm proximally before incising the circular muscle. Gradual separation was continued for 1.5–2.0 cm, followed by incision of the longitudinal muscle. The rectal muscle cuff was separated layer by layer for approximately 3–5 cm, and the circular muscle cuff was transected to the pelvic floor peritoneum. The proximal bowel was then extracted, the diseased bowel segment was resected, and the normal colon was anastomosed to the dentate line. An anal tube was left in place for 5–7 days postoperatively.

## 3. Collected data

1. **General Information**: Gender, age, reason for initial consultation, type of HSCR, preoperative Hirschsprung-associated enterocolitis (HAEC), and whether a stoma was present preoperatively.

2. **Perioperative Data**: Presence of preoperative anemia (diagnostic criteria: hemoglobin <100 g/L, or <140 g/L in neonates), preoperative hypoalbuminemia (diagnostic criteria: albumin <35 g/L), age at surgery, surgery duration, intraoperative blood loss, and postoperative hospital stay.

3. **Early Postoperative Complications**: Wound infection, HAEC, anastomotic leak, anastomotic stricture, muscle cuff infection, adhesive bowel obstruction, and rectal retraction.

## 4. Assessment of early postoperative complications

Based on the occurrence of early postoperative complications (within 30 days), patients were categorized into complication and non-complication groups. The severity of complications was assessed using the Clavien-Dindo (CD) classification system [5], which grades postoperative events according to the level of intervention required. Grade I refers to deviations from the normal postoperative course without the need for pharmacologic treatment or invasive intervention. Grade II involves the use of pharmacologic treatments beyond antiemetics or antipyretics, including transfusions and total parenteral nutrition. Grade III complications require surgical, endoscopic, or radiological intervention—with IIIa not requiring general anesthesia, and IIIb requiring general anesthesia. Grade IV includes life-threatening complications requiring intensive care, further divided into IVa (single-organ failure) and IVb (multi-organ failure). Grade V denotes death. Complications not directly related to surgery, such as unrelated systemic events, were excluded from the classification.

## 5. Statistical methods

Data collection was performed using Microsoft Excel, and statistical analysis was conducted using SPSS 29.0 software. Normally distributed continuous data were expressed as mean ± standard deviation (±s), while non-normally distributed

data were expressed as median (M) and interquartile range (IQR). Categorical data were presented as counts (percentages). Univariate analysis was conducted to identify potential factors influencing the occurrence of postoperative complications. Variables with statistically significant differences were further analyzed by multivariate logistic regression. A P value of <0.05 was considered statistically significant.

## Results

### 1. Clinical data and complications

A total of 112 children with Hirschsprung's disease (HSCR) underwent laparoscopic-assisted modified Soave procedures between January 2010 and December 2020. Of these, 83 (74.1%) were male and 29 (25.9%) were female. Constipation was the most common presenting symptom (75.9%), followed by neonatal intestinal obstruction (17.9%) and Hirschsprung-associated enterocolitis (HAEC) (6.3%). Fifty-eight patients (51.8%) had short-segment HSCR and 54 (48.2%) had long-segment disease. Preoperative stoma was performed in 15 patients (13.4%), and 19 (17.0%) had a history of HAEC. The median age at surgery was 6 months, the median operative time was 191 minutes, and the mean postoperative stay was 12.4 ± 5.3 days (Table 1).

Postoperative complications occurred in 18 patients (16.1%), involving 21 events; three patients experienced two complications each. Six patients (5.4%) required reoperation due to Clavien-Dindo grade IIIb or higher complications, including adhesive bowel obstruction, HAEC with perforation, and rectal retraction. According to the Clavien-Dindo classification, 21 events were classified as Grade I (13.6%), Grade II (50.0%), Grade IIIb (33.3%), and Grade IVa (4.5%) (Fig 1, Table 2).

**Table 1. Clinical data of 112 patients with Hirschsprung's disease (HSCR).**

| Clinical Data | Short-segment (n = 58) | Long-segment (n = 54) | Total (n = 112) |
|---|---|---|---|
| Sex (M/F) | 38/20 | 45/9 | 83/29 |
| Constipation | 45 (77.6%) | 40 (74.1%) | 85 (75.9%) |
| Neonatal intestinal obstruction | 7 (12.1%) | 13 (24.1%) | 20 (17.9%) |
| Preoperative HAEC | 9 (15.5%) | 10 (18.5%) | 19 (17.0%) |
| Preoperative stoma | 5 (8.6%) | 10 (18.5%) | 15 (13.4%) |
| Age at surgery (M, IQR) | 5 (16) | 7 (19) | 6 (18) |
| Operative time (min, IQR) | 180 (102) | 204 (117) | 191 (113) |
| Blood loss (ml, IQR) | 10 (12) | 10 (15) | 10 (15) |
| Postoperative stay (d, ± SD) | 12.2 ± 4.7 | 12.6 ± 6.0 | 12.4 ± 5.3 |
| Complication rate | 9 (15.5%) | 9 (16.7%) | 18 (16.1%) |

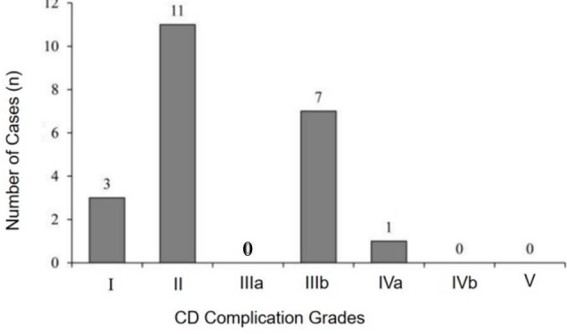

**Fig 1. Distribution of early postoperative complications according to Clavien-Dindo classification.**

**Table 2. Early postoperative complications and Clavien-Dindo Classification in 18 HSCR patients.**

| Clavien-Dindo Grade | Type of Complication | Management | Cases (%, n=21) | Reoperation (Yes/No) |
|---|---|---|---|---|
| Grade I | Perianal skin erosion | Wound care with cleaning and keeping the area dry | 2 (9.5) | No |
| | Abdominal wound infection | Treated with third-generation cephalosporin and metronidazole | 1 (4.8) | No |
| Grade II | Abdominal cavity infection | Treated with third-generation cephalosporin and metronidazole | 1 (4.8) | No |
| | HAEC | Retention enema with metronidazole and antibiotics (third-generation cephalosporin+metronidazole) | 10 (47.6) | No |
| Grad IIIb | HAEC with intestinal perforation | Emergency stoma creation, perforation repair, and elective stoma closure | 2 (9.5) | Yes |
| | Umbilical wound dehiscence with omental exposure | Emergency omental reduction and wound closure under general anesthesia | 1 (4.8) | Yes |
| | Adhesive bowel obstruction | Conservative treatment failed, requiring adhesiolysis | 3 (14.3) | Yes |
| | Rectal retraction | Emergency ileostomy followed by delayed colonic pull-through and anastomosis | 1 (4.8) | Yes |

## 2. Comparison with representative studies

Complication rates reported in representative studies of laparoscopic-assisted modified Soave procedures varied from 3.3% to 37%, depending on cohort size, follow-up duration, and classification methods. The complication rate in the present study (16.1%) was consistent with previously published data (Table 3).

## 3. Risk factor analysis

Univariate analysis showed that preoperative stoma (P=0.029), anemia (P=0.025), and hypoalbuminemia (P<0.001) were significantly associated with early postoperative complications (Table 4). Multivariate logistic regression identified preoperative stoma (OR=4.826, 95% CI=1.187–20.162, P=0.028) and hypoalbuminemia (OR=9.14, 95% CI=2.678–30.972, P<0.001) as independent risk factors, whereas anemia was not statistically significant (Table 5).

**Table 3. Summary of representative studies on early postoperative complications after laparoscopic-assisted modified soave procedures.**

| Study | Publication Year | Sample Size | Complication Rate | Follow-up Duration | Remarks |
|---|---|---|---|---|---|
| Shangjie Xiao et al. [6] | 2016 | 35 | 37% | 24 months | CD classification not applied |
| Yuan Chen [7] | 2021 | 60 | 3.33% | 6 months | CD classification not applied |
| Quande Feng et al. [8] | 2021 | 88 | 6.82% | 2–12 months | CD classification not applied |
| Fuqiang Deng et al. [9] | 2024 | 91 | 17.07% | 6 months | CD classification not applied |
| Weiyu Chen et al. [10] | 2025 | 60 | 5.0% | 1 month | CD classification not applied |
| Beltman et al. [11] | 2021 | 119 | 21.0% | 6 months | CD classification applied (transanal approach) |
| Ahmad et al. [12] | 2022 | 103 | 19.4% | 12 months | Focused on postoperative failure and management |
| Gershon et al. [13] | 2023 | 85 | 24.7% | 1–3 years | Systematic evaluation of HAEC and outcomes |
| Present study | 2025 | 112 | 16.1% | 1 month | CD classification applied (laparoscopic-assisted Soave) |

CD=Clavien-Dindo; HAEC=Hirschsprung-associated enterocolitis.

**Table 4. Univariate analysis of factors influencing early postoperative complications in 112 HSCR patients.**

| Factors | Complication Group (n = 19) | Non-Complication Group (n = 93) | $\chi^2$ Value | P Value |
|---|---|---|---|---|
| Gender | | | 0.002 | 0.809 |
| Male | 14 | 69 | | |
| Female | 5 | 24 | | |
| Age at surgery (months) | | | 0.201 | 0.453 |
| ≤3 | 3 | 22 | | |
| >3 | 16 | 71 | | |
| Clinical type | | | 0.179 | 0.672 |
| short segment disease | 9 | 49 | | |
| long segment disease | 10 | 44 | | |
| Preoperative HAEC | | | 0.034 | 0.853 |
| Yes | 4 | 15 | | |
| No | 15 | 78 | | |
| Preoperative stoma | | | 4.773 | 0.029 |
| Yes | 6 | 9 | | |
| No | 13 | 84 | | |
| Preoperative anemia | | | 5.029 | 0.025 |
| Yes | 8 | 15 | | |
| No | 11 | 78 | | |
| Preoperative hypoalbuminemia | | | 17.728 | <0.001 |
| Yes | 10 | 9 | | |
| No | 9 | 84 | | |
| Duration of surgery (hours) | | | 2.556 | 0.110 |
| ≤3 | 5 | 43 | | |
| >3 | 14 | 50 | | |

**Table 5. Multivariate logistic regression analysis of factors influencing early postoperative complications in 112 HSCR patients.**

| Factors | Regression Coefficient | Standard Error | Wald Value | P Value | ORValue | 95%CI | |
|---|---|---|---|---|---|---|---|
| | | | | | | Lower limit | Upper limit |
| Preoperative stoma | 1.587 | 0.723 | 4.826 | 0.028 | 4.891 | 1.187 | 20.162 |
| Preoperative anemia | 1.205 | 0.641 | 3.532 | 0.06 | 3.335 | 0.95 | 11.714 |
| Preoperative hypoalbuminemia | 2.213 | 0.623 | 12.626 | <0.001 | 9.14 | 2.697 | 30.972 |

## Discussion

Early postoperative complications are an important measure of surgical safety in Hirschsprung's disease (HSCR). Reported incidences vary widely across institutions, largely due to inconsistent definitions and grading systems [14]. This study systematically applied the Clavien–Dindo (CD) classification to early postoperative complications following laparoscopic-assisted modified Soave procedures in children, providing a structured and severity-based framework for outcome assessment [15,16].

Nearly one-third of complications were classified as CD grade IIIb or higher, all requiring surgical intervention under general anesthesia. Although less frequent than grade I–II events, these high-grade complications had substantial clinical impact, resulting in prolonged hospitalization and repeat surgery [17]. Such findings highlight the importance of early

recognition and prompt intervention in managing severe complications, consistent with observations from pediatric cohorts that employed the CD framework [18,19].

HAEC was the most common complication in this study, accounting for nearly half of all events. Most cases were treated conservatively (CD grade II), but two children developed intestinal perforation (CD grade IIIb), underscoring the potential for rapid progression to life-threatening conditions. The CD classification complements existing diagnostic criteria by grading outcomes according to treatment intensity, thereby providing a reproducible metric for surgical benchmarking [20–22].

Rectal retraction, although observed in only one patient, represented another severe complication (CD grade IIIb) that required reoperation. This highlights that even rare complications can substantially affect recovery and reinforces the need for meticulous intraoperative mobilization and anastomotic tension control to minimize risk.

Multivariate analysis demonstrated that preoperative hypoalbuminemia and stoma formation were independent risk factors for early postoperative complications. Hypoalbuminemia has been associated with adverse outcomes in general and colorectal surgery, and our results extend this association to HSCR-specific morbidity. This finding suggests that targeted nutritional optimization before surgery may help reduce postoperative risk. Similarly, children with preoperative stomas showed higher complication rates, which may reflect both baseline disease severity and stoma-related nutritional or metabolic imbalances. These results emphasize that perioperative outcomes are shaped not only by surgical technique but also by preoperative patient condition.

The present study enrolled 112 patients over a 10-year period, providing robust data to examine complication patterns across HSCR subtypes. Although complication rates did not differ significantly between short- and long-segment HSCR, children with long-segment disease presented with greater clinical complexity, suggesting that disease extent may influence perioperative management needs even if overall frequencies appear similar.

When contextualized with domestic and international reports, the complication rate observed here (16.1%) is broadly consistent with published findings, especially in studies that adopted structured grading systems such as Clavien–Dindo [15,16,23,24]. This underscores the value of standardized reporting systems in enabling cross-cohort comparability and supporting methodological alignment in pediatric surgical research.

## Limitations

This study was a retrospective, single-center analysis with a 30-day postoperative window, which may have introduced reporting bias and limits generalizability. The absence of long-term functional outcomes, such as recurrent constipation and soiling, further constrains interpretation. Although the cohort comprised 112 children, the sample size remains limited relative to multicenter datasets. Future multicenter prospective studies with standardized reporting and extended follow-up are needed to validate and expand upon these findings.

## Conclusion

Early postoperative complications following laparoscopic-assisted modified Soave procedures are not uncommon, and one-third required surgical reintervention. Preoperative hypoalbuminemia and stoma creation emerged as independent risk factors, highlighting the importance of nutritional optimization and careful perioperative risk stratification. Adoption of the Clavien–Dindo system offers a reproducible framework for complication grading, promoting transparency, benchmarking, and quality improvement in pediatric surgical care.

## Supporting information

**S1 File. Raw data of clinical variables and outcomes used for statistical analysis in this study.**
(ZIP)

## Acknowledgments

The authors would like to thank the pediatric surgery team, nursing staff, and medical record department of Zunyi Medical University Affiliated Hospital for their valuable assistance in data collection and patient care. We are also grateful to the patients and their families for their cooperation and trust throughout the study period.

## Author contributions

**Conceptualization:** Xianhui Shang, Renbiao Huang, Hao Fu.

**Data curation:** Yuanmei Liu, Luping Xiang.

**Formal analysis:** Hao Fu, Yan Qu.

**Investigation:** Renbiao Huang, Shiyu Xu.

**Methodology:** Xianhui Shang, Yuanmei Liu.

**Resources:** Yan Qu.

**Software:** Renbiao Huang, Luping Xiang, Yan Qu.

**Supervision:** Xianhui Shang, Yuanmei Liu.

**Validation:** Yuanmei Liu.

**Visualization:** Hao Fu, Shiyu Xu.

**Writing – original draft:** Renbiao Huang.

**Writing – review & editing:** Xianhui Shang, Yuanmei Liu, Hao Fu, Luping Xiang, Shiyu Xu.

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
