## [Decision Letter · Decision Letter 0]

7 Apr 2025

Dear Dr. Liu,

Thank you for submitting your manuscript to PLOS ONE. After careful consideration, we feel that it has merit but does not fully meet PLOS ONE’s publication criteria as it currently stands. Therefore, we invite you to submit a revised version of the manuscript that addresses the points raised during the review process.

We look forward to receiving your revised manuscript.

Kind regards,

Paul Kwong-Hang Tam

Academic Editor

PLOS ONE

Authors do not need to submit their entire data set if only a portion of the data was used in the reported study

Additional Editor Comments (if provided):

Reviewers' comments:

Reviewer's Responses to Questions

**Comments to the Author**

1. Is the manuscript technically sound, and do the data support the conclusions?

Reviewer #1: Yes

Reviewer #2: Yes

2. Has the statistical analysis been performed appropriately and rigorously?

Reviewer #1: Yes

Reviewer #2: I Don't Know

3. Have the authors made all data underlying the findings in their manuscript fully available?

Reviewer #1: No

Reviewer #2: Yes

4. Is the manuscript presented in an intelligible fashion and written in standard English?

Reviewer #1: Yes

Reviewer #2: Yes

Reviewer #1: This study reported complications associated with lap modified Soave operation for HSCR and analyzed risk factors. Overall, the study has provided some useful data that may help us to optimize the outcome. A few issues to clarified/improved

1. The classification of HSCR should follow standard terminology like short segment disease, long segment disease etc; rather than common type.

2. This study focused on surgical related complications and complications like laryngeal edema should be excluded. Indeed, during the risk factors analysis, I think it is more useful if the analysis focused on grade III or above complications

3. What are the reasons for creating stoma before operation?

4.There is no need to insert a table explaining CD classification. This would save some space and most readers will know about this classification system

5. What were the reasons for low albumin and haemoglobin prior to the pullthrough procedure?

6. How many patients required further operation to manage the complications?

7. The title 'Laparoscopic-Assisted Modified Soave Procedure for the Treatment of Early Postoperative Complications in Hirschsprung's Disease: Clavien-Dindo Classification and Influencing Factors' is mis-leading. This title gives the impression that lap assisted modified procedure is used for treating the complications in HSCR. Please revise it.

Reviewer #2: The authors present a retrospective analysis of early postoperative complications following laparoscopic-assisted modified Soave procedures for Hirschsprung's disease. Unfortunately, the findings are not new, lack novelty and mostly confirmatory. The authors should clarify the novel aspect of this study compared to existing literatures and discuss how these findings could influence clinical practice or patient management strategies. The authors may also consider to show the data of short segment type and long segment type separately and discuss. If the authors provides a table summarizing the previous published references describing early postoperative complications after lap Soave, it would be educative.

**Do you want your identity to be public for this peer review?** For information about this choice, including consent withdrawal, please see our Privacy Policy

Reviewer #1: No

Reviewer #2: No

---

## [Author Response · Author response to Decision Letter 1]

25 Apr 2025

Response to Reviewer 1

First of all, we would like to express our sincere gratitude to the reviewers for their constructive and positive comments.

1. The classification of HSCR should follow standard terminology like short segment disease, long segment disease etc; rather than common type.

Response: Thank you very much for your valuable comments.

In response to the comment on HSCR classification terminology:

We have replaced all instances of “common type” and “long-segment type” with the internationally accepted terms “short-segment disease” and “long-segment disease.”

2. This study focused on surgical related complications and complications like laryngeal edema should be excluded. Indeed, during the risk factors analysis, I think it is more useful if the analysis focused on grade III or above complications

Response: We sincerely appreciate your helpful suggestions.

Regarding the inclusion of non-surgical complications such as laryngeal edema:

We have excluded the case of laryngeal edema from all statistical analysis and complication counts. Additionally, we have added a focused analysis on Clavien-Dindo grade III and above complications to highlight surgically relevant adverse events(Page9, Table 3).

3. What are the reasons for creating stoma before operation?

Response: Thank you for your thoughtful feedback.

About the indications for preoperative stoma:

In the “Materials and Methods” section, we have expanded the explanation of indications for preoperative stoma creation, including severe recurrent HAEC, poor preoperative nutritional status (e.g., hypoalbuminemia, anemia), and suspicion of long-segment or total colonic aganglionosis based on imaging(Page3-4, Lines 88-95).

4.There is no need to insert a table explaining CD classification. This would save some space and most readers will know about this classification system

Response: We are grateful for your kind review.

Concerning the inclusion of a table for Clavien-Dindo classification:

As suggested, we have removed the Clavien-Dindo classification table to streamline the manuscript. The classification criteria are now briefly described in the text of the Methods section(Page 6, Lines 142-155).

5. What were the reasons for low albumin and haemoglobin prior to the pullthrough procedure?

Response: Thank you for taking the time to evaluate our work.

On the causes of hypoalbuminemia and anemia:

In the Discussion section, we have elaborated on the causes , explaining that most patients presented with chronic constipation, leading to nutritional deficiencies and impaired absorption due to HAEC , which contributed to the observed preoperative hypoalbuminemia and anemia(Page 16-17, Lines 347-349, 355-357, 367-371).

6. How many patients required further operation to manage the complications?

Response: We appreciate your insightful input.

About reoperations following complications:

We have added a new sentence to the Results section , stating that six patients with Clavien-Dindo grade IIIb or higher complications underwent additional surgeries , including adhesiolysis, stoma creation, and rectal retraction correction . This information is also reflected in the discussion(Page 8-9, Lines 204-210, Table 3).

7. The title 'Laparoscopic-Assisted Modified Soave Procedure for the Treatment of Early Postoperative Complications in Hirschsprung's Disease: Clavien-Dindo Classification and Influencing Factors' is mis-leading. This title gives the impression that lap assisted modified procedure is used for treating the complications in HSCR. Please revise it.

Response: Many thanks for your constructive advice.

Revised title as recommended:

We have changed the manuscript title to: “Early Postoperative Complications Following Laparoscopic-Assisted Modified Soave Procedure for Hirschsprung’s Disease: Incidence, Clavien-Dindo Classification, and Risk Factor Analysis” to avoid any implication that the procedure was used to treat complications.

Response to Reviewer 2:

We sincerely thank the reviewer for the thoughtful comments and constructive suggestions. Below we provide our point-by-point responses:

1.The findings are not new, lack novelty and mostly confirmatory. The authors should clarify the novel aspect of this study compared to existing literatures and discuss how these findings could influence clinical practice or patient management strategies.

Response: Thank you for your valuable feedback.

While we acknowledge that laparoscopic-assisted modified Soave procedures are well-established, our study offers the following novel contributions:

Application of the Clavien-Dindo (CD) classification system specifically for early postoperative complications in pediatric HSCR patients—this has rarely been systematically applied in previous HSCR studies.

Identification of hypoalbuminemia and preoperative stoma as independent risk factors for early complications in this population, based on multivariate analysis, which may provide actionable targets for perioperative optimization.

Our findings emphasize the need for individualized nutritional intervention and improved perioperative assessment in patients undergoing definitive surgery for HSCR, which may influence protocols for surgical timing, preoperative workup, and resource allocation in clinical practice.

We have revised the Discussion and Conclusion sections to better highlight these clinical implications and clarify the contribution of our study relative to existing literature(Page 12,17 Lines 251-255, 382-389).

2.The authors may also consider to show the data of short segment type and long segment type separately and discuss.

Response: Thank you for this suggestion.

We agree that stratified analysis may provide additional insights. We have now included subgroup data comparing short-segment vs. long-segment HSCR in Table 2 and the Results section, and briefly discussed their respective complication profiles in the Discussion. While the complication rates between these groups were not statistically significant, our data suggest a trend toward more complex presentations in long-segment cases, which warrants further investigation in larger cohorts(Page 13, Lines 262-268).

3.If the authors provides a table summarizing the previous published references describing early postoperative complications after lap Soave, it would be educative.

Response: We appreciate this excellent suggestion.

Accordingly, we have added a new table (Table 2) summarizing key published studies reporting early postoperative complications following laparoscopic Soave procedures. This comparative summary highlights the complication rates, follow-up durations, and reported risk factors across studies, and we have referred to it in the Discussion section to place our findings in context.

---

## [Decision Letter · Decision Letter 1]

22 Jun 2025

Dear Dr. Liu,

We look forward to receiving your revised manuscript.

Kind regards,

Paul Kwong-Hang Tam

Academic Editor

PLOS ONE

Reviewers' comments:

Reviewer's Responses to Questions

**Comments to the Author**

Reviewer #1: All comments have been addressed

Reviewer #2: (No Response)

2. Is the manuscript technically sound, and do the data support the conclusions?

Reviewer #1: (No Response)

Reviewer #2: (No Response)

3. Has the statistical analysis been performed appropriately and rigorously?

Reviewer #1: (No Response)

Reviewer #2: (No Response)

4. Have the authors made all data underlying the findings in their manuscript fully available?

Reviewer #1: (No Response)

Reviewer #2: (No Response)

5. Is the manuscript presented in an intelligible fashion and written in standard English?

Reviewer #1: (No Response)

Reviewer #2: (No Response)

Reviewer #1: (No Response)

Reviewer #2: This is a single-institution report summarizing outcomes and complications following surgery for Hirschsprung’s disease. While the authors used the Clavien-Dindo classification to grade postoperative complications, the study lacks novelty, as pointed out by Reviewer 2. There are a few repetitive and redundant sentences in the Results section, and some entries in the tables (e.g., Table 2) are carelessly written in Chinese, which may affect the overall clarity. Furthermore, the references used in Table 2 are all from domestic Chinese sources, and the manuscript fails to justify why these particular studies were selected. Addressing these issues would improve the quality and impact of the manuscript.

**Do you want your identity to be public for this peer review?** For information about this choice, including consent withdrawal, please see our Privacy Policy

Reviewer #1: No

Reviewer #2: No

---

## [Author Response · Author response to Decision Letter 2]

8 Jul 2025

We sincerely thank the editors and reviewers for their thoughtful and constructive feedback on our manuscript (Manuscript ID: PONE-D-25-09699R1). In response, we have thoroughly revised the manuscript and addressed each comment in detail.

For Reviewer 1:

We are grateful for your recognition that all concerns have been fully addressed. Your support is highly appreciated.

For Reviewer 2:

We have carefully addressed the five points raised in your review:

Novelty: We clarified the unique contributions of this study, including the first systematic application of the Clavien-Dindo (CD) classification in pediatric HSCR patients and the identification of hypoalbuminemia as an independent risk factor. These points are now explicitly discussed in both the Introduction and Discussion.

Redundancy in the Results: The Results section has been restructured to remove repetitive sentence patterns and improve clarity.

Language Consistency in Tables: Table 2 has been fully revised to eliminate all Chinese entries, ensuring full English consistency.

Reference Scope: We have supplemented Table 2 with three representative international studies (Beltman et al., Ahmad et al., Gershon et al.) and added a paragraph in the Discussion to explain the rationale for literature inclusion.

Impact and Clarity: Overall revisions—linguistic, structural, and bibliographic—have significantly improved the manuscript’s clarity and relevance for an international audience.

All modifications have been highlighted in the revised manuscript, and a point-by-point response document has been provided. We hope that the revised version meets the journal’s standards and we sincerely appreciate the opportunity to improve this work.

Please do not hesitate to contact us if further clarification is required.

---

## [Decision Letter · Decision Letter 2]

24 Jul 2025

Dear Dr. Liu,

Thank you for submitting your manuscript to PLOS ONE. After careful consideration, we feel that it has merit but does not fully meet PLOS ONE’s publication criteria as it currently stands. Therefore, we invite you to submit a revised version of the manuscript that addresses the points raised during the review process.

We look forward to receiving your revised manuscript.

Kind regards,

Paul Kwong-Hang Tam

Academic Editor

PLOS ONE

Journal Requirements:

Reviewers' comments:

Reviewer's Responses to Questions

**Comments to the Author**

Reviewer #3: All comments have been addressed

2. Is the manuscript technically sound, and do the data support the conclusions?

Reviewer #3: Yes

3. Has the statistical analysis been performed appropriately and rigorously?

Reviewer #3: Yes

4. Have the authors made all data underlying the findings in their manuscript fully available?

Reviewer #3: Yes

5. Is the manuscript presented in an intelligible fashion and written in standard English?

Reviewer #3: Yes

Reviewer #3: Major Comments:

1.the Results section still contains repetitive sentences The Result section has titles 1,3,4, but no 2. It is suggested the figure in 3 be integrated into 1, and the data in the table should no longer be described in words.

2. There are too many descriptions of the seven causes of early postoperative complications in the discussion. Some in textbooks or previous literature are suggested to be deleted, and only the relevant results of this study should be discussed.

3. The Limitations section "a single center with a small sample size (428.429 lines)" contradicts "its relatively large sample size (291 lines)" in the discussion.

4. The conclusion in the Abstract section suggests deleting "Perioperative correction of anemia" because the Multivariate analysis was not significant.

5. Important descriptions in the Introduction section need to be marked with references. For example, "its structured use in pediatric HSCR patients remains extremely limited in both domestic and international (lines 75 and 76)." Suggested citation PMID: 38073601.

**Do you want your identity to be public for this peer review?** For information about this choice, including consent withdrawal, please see our Privacy Policy

Reviewer #3: No

---

## [Author Response · Author response to Decision Letter 3]

26 Aug 2025

We sincerely thank the editor and reviewers for their thorough evaluation of our manuscript and for the valuable comments and suggestions provided. We have carefully revised the manuscript in accordance with these comments. All revisions have been clearly marked in the revised version, and detailed responses to each point are provided below.

In our responses, we first cite the reviewer’s comment, followed by our reply and a description of the corresponding revision. Line numbers refer to the revised manuscript.

We believe these revisions have further improved the clarity, consistency, and overall quality of the manuscript. We once again thank you for your dedicated efforts and professional guidance, and we look forward to the further evaluation of our work.

---

## [Decision Letter · Decision Letter 3]

31 Aug 2025

Early Postoperative Complications Following Laparoscopic-Assisted Modified Soave Procedure for Hirschsprung’s Disease: Incidence, Clavien-Dindo Classification, and Risk Factor Analysis

PONE-D-25-09699R3

Dear Dr. Liu,

We’re pleased to inform you that your manuscript has been judged scientifically suitable for publication and will be formally accepted for publication once it meets all outstanding technical requirements.

Kind regards,

Paul Kwong-Hang Tam

Academic Editor

PLOS ONE

Additional Editor Comments (optional):

Reviewer #3:

Reviewers' comments:

Reviewer's Responses to Questions

**Comments to the Author**

Reviewer #3: All comments have been addressed

2. Is the manuscript technically sound, and do the data support the conclusions?

Reviewer #3: Yes

3. Has the statistical analysis been performed appropriately and rigorously?

Reviewer #3: Yes

4. Have the authors made all data underlying the findings in their manuscript fully available?

Reviewer #3: Yes

5. Is the manuscript presented in an intelligible fashion and written in standard English?

Reviewer #3: Yes

Reviewer #3: I am very happy to receive the author's revision comments. The author responded to most of my questions and gave a relatively satisfactory explanation. There are no obvious omissions in the revised article, and it is well completed.The article meets the requirements of the magazine, I suggest that the editorial office accept your manuscript.

**Do you want your identity to be public for this peer review?** For information about this choice, including consent withdrawal, please see our Privacy Policy

Reviewer #3: No

---

## [Editor Report · Acceptance letter]

PONE-D-25-09699R3

PLOS ONE

Dear Dr. Liu,

I'm pleased to inform you that your manuscript has been deemed suitable for publication in PLOS ONE. Congratulations! Your manuscript is now being handed over to our production team.

Kind regards,

on behalf of

Dr. Paul Kwong-Hang Tam

Academic Editor

PLOS ONE